# Calculation of the Rub' al Khali Sand Dune Volume for Estimating Potential Sand Sources

**Fahad Almutlaq** [1,†], **Faten Nahas** [1,*,†] **and Kevin Mulligan** [2]

1  Geography Department, College of Arts, King Saud University, Riyadh 11451, Saudi Arabia; falmutlaq@ksu.edu.sa
2  Department of Geosciences, Texas Tech University, Lubbock, TX 79409, USA; kevin.mulligan@ttu.edu
*  Correspondence: fnahas@ksu.edu.sa
†  First joint author.

**Abstract:** The Rub' al Khali desert (or Empty Quarter) is the largest and perhaps most significant sand sea in the world. Located on the southern Arabian Peninsula, the dune field has remained largely unexplored owing to the harsh clime and difficult terrain. This study takes advantage of geospatial technology (interpolations, supervised classification, minimum focal statistic) to extract information from the data contained in global Digital Elevation Model (DEM)s, satellite imagery. The main objective here is to identify and map different dune forms within the sand sea, estimate the volume of sand and explore probable sources of sand. The analysis of dune color strongly suggests that the sand is not completely reworked and intermixed. If this is true, a spatial variability map of the mineral composition of the sand could be very revealing. The red sand is quite pronounced, the largest volume of sand (~36%) is associated with the yellow color class. Yellow sand covers most of the western part of the dunes field and seems to be a transitional color between red and white sand in the eastern part of the dune field. This suggests that the yellow sand might be derived from both local and regional sources, or it might be less oxidized, reworked, or have a different composition that represents a combination of red and white sand.

**Keywords:** Rub' al Khali; DEM; sand; volume; sand sources; dune; wadi

## 1. Introduction

A question has always existed as to the reason why such a huge amount of quartz sand exists in the Rub' al Khali, while the surrounding geology consists largely of carbonates [1]. This observation raises interesting questions with regard to the origin and age of the dunes. Given the eastward orientation and slope of the Rub' al Khali basin and the alignment of dunes with respect to dominant northwest winds, there is a strong consensus that the sand dunes have developed from both local and regional sources [1,2]. In this regard, [3] was the first to suggest that wadis flowing eastward from the Arabian Shield mountains contribute local sand to the western part of the dune field. More importantly, [4,5] recognized that the Arabian Gulf was dry during the last glacial maximum (17,000–25,000 BP) when sea level was about 120 m lower than today. With the dry Arabian Gulf exposed as a source, winds from the northwest could transport an enormous volume of sand over a period of thousands of years. To investigate the potential sources of sand and estimate the total sand volume in the Rub' al Khali based on sand dune color, field investigation, laboratory analysis of sediments, and visual interpretation of satellite data are required to be considered.

To date, there does not appear to be any research undertaken to classify satellite imagery based on the spectral signature of the dune sand. Yet, these color differences are evident in satellite imagery. In this regard, the color of the sand surface is important because it should be related to the composition and/or age of the sand, both of which are relevant to understanding the potential sources.

Recognizing that the dunes within the Rub' al Kahli are composed of sand from several different sources then leads to the next basic question. How much sand is there in the Rub' al Khali, and what percent of this sand is derived from different sources? As our understanding of potential sources becomes more refined, future research might be able to quantify the relative volume from different sources, but for now, there does not appear to be any quantitative estimate of the total volume of sand. The objective of this study is to analyze satellite imagery to identify and map significant differences in sand dune colors. While this analysis does not attempt to identify particular sources of sand, these color differences can be used to identify different regions of the dune field that might share a common source or might be a common age.

## 2. Materials and Methods

### 2.1. Study Area

The Rub' al Khali is the largest contiguous sand sea in the world, located in the southern part of the Arabian Desert [6,7]. The Arabian Desert occupies most of the Arabian Peninsula and is located between the Arabian Gulf and Gulf of Oman to the east, the Arabian Sea (northwest Indian Ocean) and the Gulf of Aden to the south, and the Red Sea to the east. In the northern part of the Arabian Peninsula, the Arabian Desert transitions into the Syrian Desert [8] (Figure 1).

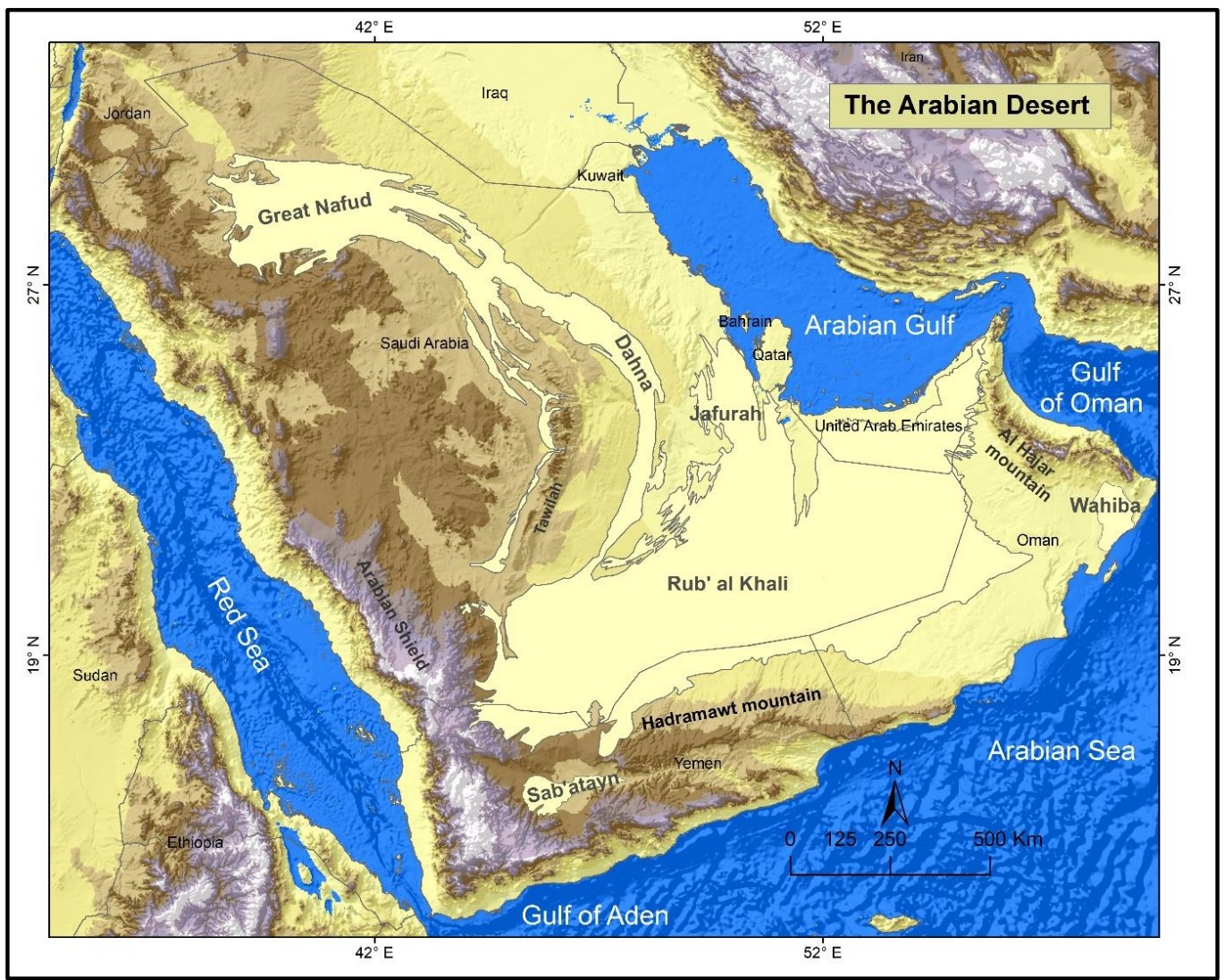

**Figure 1.** Location and regional setting of the Rub' al Khali on the Arabian Peninsula.

The figure below shows the main concept of methodology (Figure 2).

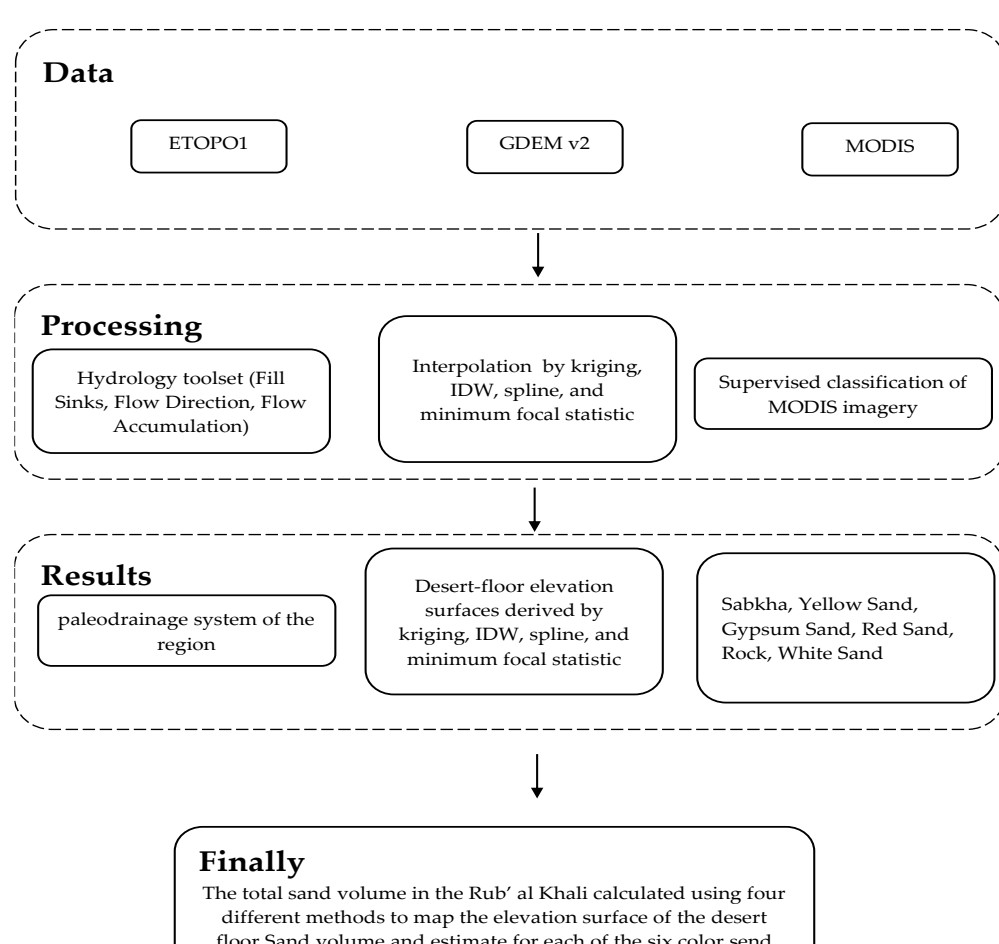

**Figure 2.** The main concept of the methodology for sand dune volume estimating potential from sand sources.

*2.2. Data*

Today there are many satellite platforms and sensors that provide images of the Earth's surface [9]. Comparing these sources, the Landsat Four MODIS data scenes were downloaded from the USGS GloVis (Global Visualization Viewer). These four scenes were then mosaicked and clipped to the study area using the boundary defined by the Saudi Geological Survey (2011) [10]. For the true color band combination, MODIS imagery has a spatial resolution of 500 m, the purpose of the analysis is to classify and map the major color difference in the sand surface across the entire dune field. While any DEM can be used to generate an artificial network of streams, ETOPO1 is the only global DEM that provides elevation data for both the land surface and the ocean floor. With a spatial resolution of 1.8 km$^1$ arc minute, ETOPO1 is also well-suited for this type of regional-scale analysis that covers the entire Arabian Peninsula. To reconstruct the regional paleo drainage system, the ETOPO1 Global Relief Model was downloaded from the NOAA, National Centers for Environmental Information using the grid-registered version of the dataset. To map the surface of the dunes and the desert floor beneath the dunes, the Advanced Spaceborne Thermal Emission and Reflection Radiometer (ASTER) GDEM v2 was chosen as the primary elevation source [11,12]. This particular DEM was selected because it provides the best high-resolution elevation data available. The ASTER dataset is a near-global DEM with a spatial resolution of 1 arc-second (~30 m) (Table 1).

**Table 1.** Imagery and Digital Elevation Models used for terrain analysis.

| Name | Original Source | Resolution | Year | Source Link |
|---|---|---|---|---|
| MODIS | the Terra and Aqua satellites | 500 m | 2016 | http://glovis.usgs.gov/ (accessd on 22 January 2022) |
| GDEM v2 | Derived by fusing ASTER GDEM v2 with SRTM v4.1 | 30 m 90 m | 2014 | http://www.earthenv.org/DEM (accessd on 22 January 2022) |
| ETOPO1 | Global Relief Model | 1.8 km | 2010 | https://www.ngdc.noaa.gov/mgg/global/ (accessd on 22 January 2022) |

*2.3. Methodology*

To better understand the potential sources of sand and estimate the total sand volume in the Rub' al Khali, the analysis in this study is subdivided into four parts. In the first part of this study, elevations are extracted from a DEM to map the elevation of the desert floor beneath the dunes. In the second part of the study, the elevation surface of the desert floor is subtracted from the elevation of the sand surface to calculate the total volume of sand in the Rub' al Khali. In the third part of the study, satellite imagery is analyzed to map the differences in the color of the sand surface, which is likely related to the composition and/or age of the sand [13]. Lastly, in the fourth part of the study, a drainage network is created from a DEM to reconstruct the paleo drainage system of the region during the last glacial maximum.

To estimate the total volume of sand in the desert, the elevation surface of the desert floor is subtracted from the elevation surface of the original DEM. As before, the ASTER GDEM v2 is used in this analysis as the primary elevation source and this dataset has a spatial resolution of 30 m (1 arc second) [14].

In the first step, to build the desert floor of Rub al Kalih, Google Earth Pro software was used. The points were then converted to a vector GIS layer and overlaid on the ASTER DEM in ArcMap. While sand covers the majority of the Rub' al Khali, the desert floor is exposed between many of the dunes, and these exposures are vividly apparent on satellite imagery. Some of these interdune flats contain gravel sheets, sabkhas, and other apparently evaporative surfaces [15]. The point coordinates were then used to extract raster-cell elevation values from the ASTER elevation model. In this case, the elevation attribute was added to each point using the Extract to Point tool in the Spatial Analyst extension in ArcGIS. These point elevation attributes represent the elevation of the desert floor (Figure 3).

Once the maps of the desert floor were developed, both the original ASTER surface elevation and each of the desert floor surfaces were projected to calculate the sand volume in meaningful units (m$^3$). In this case, the datasets were projected using an Albers Equal Area projection, with the central meridian set to 50° E and the two standard parallels set to 17.5° N and 22° N. As part of the projection process, the rasters are resampled and the original 1 arc-second (~30 m) cell size is changed to create grid cells in planer units. To minimize error, the cell size of the output raster was specified as 30 × 30 m, which closely approximates the original cell size.

With all of the datasets projected, map algebra was used to subtract each of the desert floor elevation surfaces from the original ASTER DEM, where the ASTER elevation model represents the elevation of the dune surface. To subtract the elevation surfaces, the analysis was carried out using the Raster Calculator tool in the Spatial Analyst extension in ArcGIS.

Figure 4 shows the distribution of the 700 point locations used to define the desert floor. In general, the points are well distributed over the dune field, and the large number of point locations should be adequate to define the desert floor elevation surface.

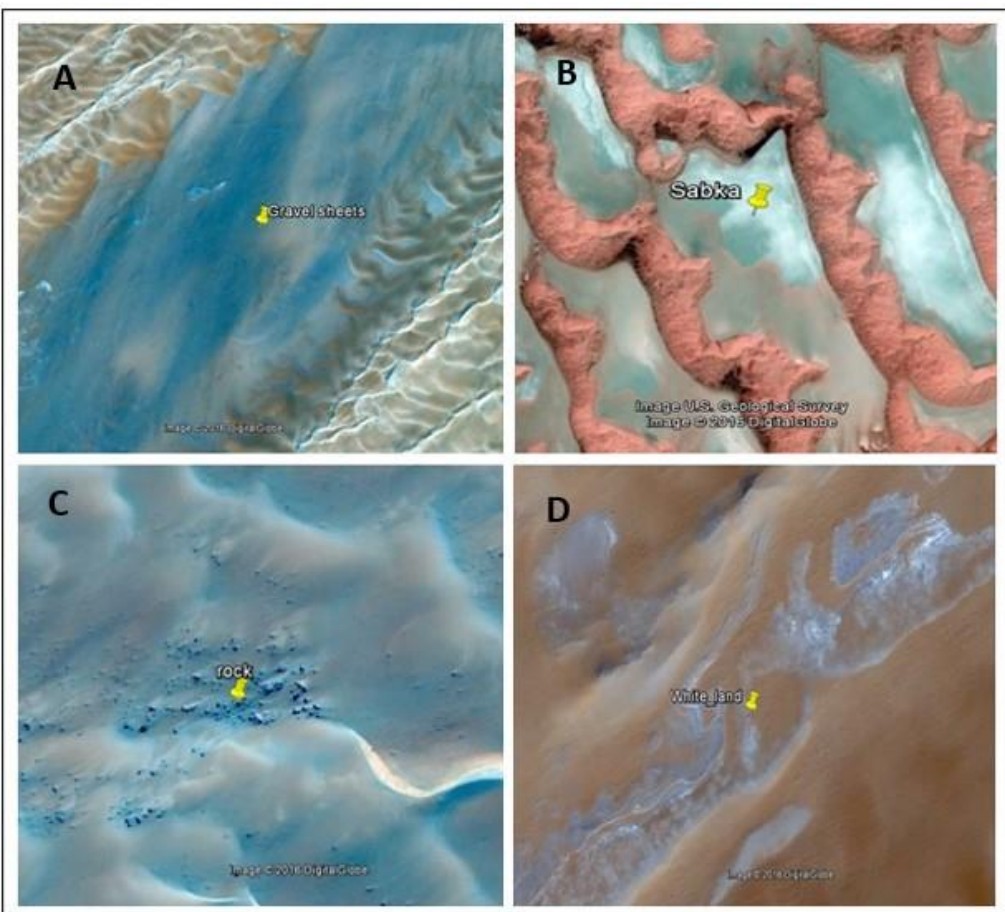

**Figure 3.** Examples of (**A**) gravel sheets, (**B**) sabkhas, (**C**) rock outcrops and (**D**) non-sand surfaces.

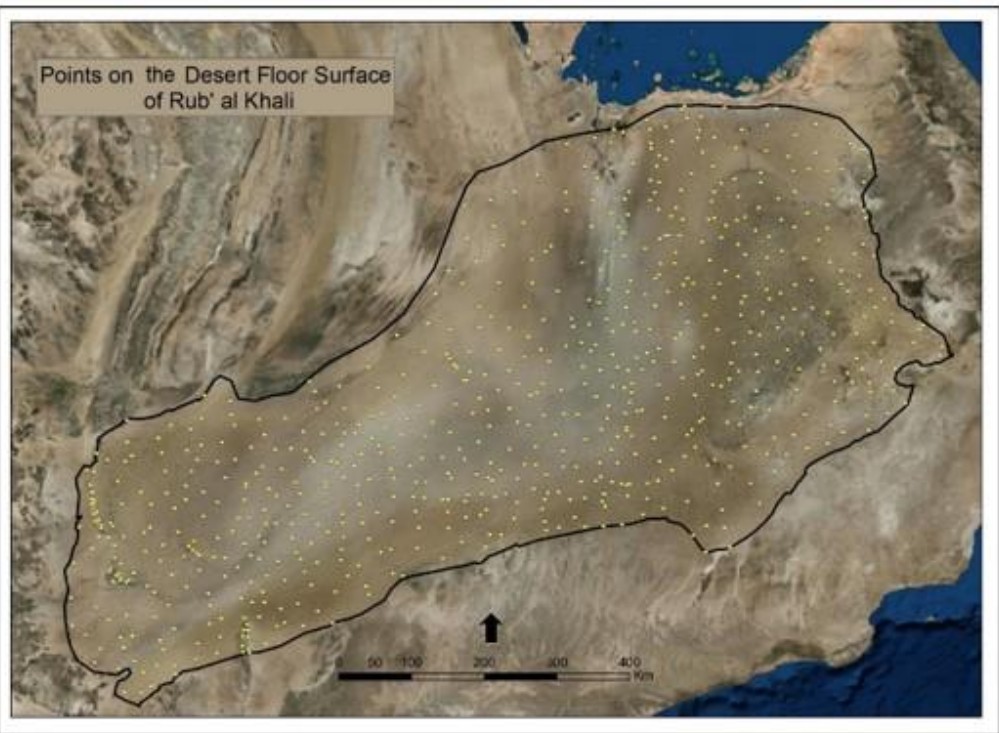

**Figure 4.** Distribution of the points used to define the elevation of the desert floor.

After the point dataset was completed, the desert floor elevation surface was created using three different interpolation techniques. Interpolation is a method or mathematical function that estimates the values at locations, in this case, the elevation values of the desert floor between the point locations. The three most common techniques of spatial interpolation are kriging, inverse distance weighting (IDW), and spline [16,17]. Each of these different interpolation methods perform well for the interpolation of geomorphologically smooth areas [18]. Kriging is an advanced interpolation technique that assumes that the distance or direction between the sample points reflects a spatial correlation that can be used to explain the variation in the surface [19]. The spline method predicts values using a mathematical function that minimizes the total surface curvature [20]. IDW assumes that the closest point values have a stronger impact on interpolated values than distant observations, resulting in a smooth surface [17].

In addition, focal statistics was used as another method to map the surface of the desert floor. In this case, the elevation of the desert floor beneath the dunes was derived more directly by extracting the minimum elevation (minimum focal statistic) in a focal block (neighborhood) containing 30 × 30 grid cells (~900 × 900 m). This approach assumes that the neighborhood is large enough to capture the elevation of at least one grid cell representing the surface of the desert floor. While this approach should be valid for most of the dune field where exposed surfaces are obvious, in some parts of the dune field, the minimum elevation might reflect older, indurated, paleo-sand surfaces lying beneath the modern dunes.

In the second step, the results from this analysis produced four raster products as output representing the elevation difference between the dune surface and the desert floor: one for the kriged surface, one for the IDW surface, one for the spline surface, and one for the focal statistic surface [21]. The elevation-difference values were then multiplied by the area of a grid cell (900 m$^2$) to calculate the volume for each grid cell (Figure 5). The grid cell volume values were then summed to calculate the total sand volume where:

$$\text{Total Volume} = \sum\{(\text{dune surface elevation} - \text{desert floor elevation}) \times \text{grid cell area}\} \tag{1}$$

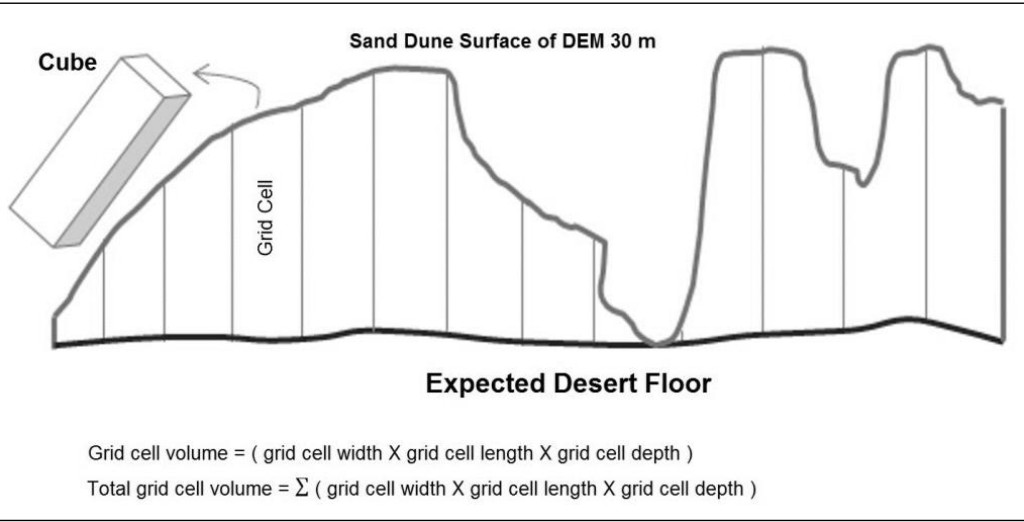

**Figure 5.** The method used to calculate the sand volume in the Rub' al Khali desert.

In the third step, to map differences in the spectral reflectance of the desert surface, the image processing tools in ERDAS IMAGINE were used to develop a supervised classification. This supervised classification was designed to map the spatial variability in the spectral reflectance of the true color bands [22,23]. The goal in this step is to calculate the volume of sand represented by each class of color. In MODIS imagery, these true-color

bands include band 1 (red), band 4 (green), and band 3 (blue) (Figure 6). To develop the supervised classification, six training sites were identified based mainly on differences in color. These six training sites were named: (1) gypsum sand, (2) sabkha, (3) yellow sand, (4) red sand, (5) white sand, and (6) rock sand (rock with sand). Given that the purpose of the analysis is to classify and map the major color difference in the sand surface across the entire dune field, a lower resolution image source is more suitable. For this analysis, the imagery captured by the MODIS (Moderate Resolution Imaging Spectroradiometer) instrument onboard the Terra and Aqua satellites is well suited [24]. For the true color band combination, MODIS imagery has a spatial resolution of 500 m. Once the image was prepared for analysis, the image processing tools in ERDAS IMAGINE were used to conduct a supervised classification. Thus, it is important to recognize that the spectral reflectance of the surface can also be related to the composition and color of interdune flats.

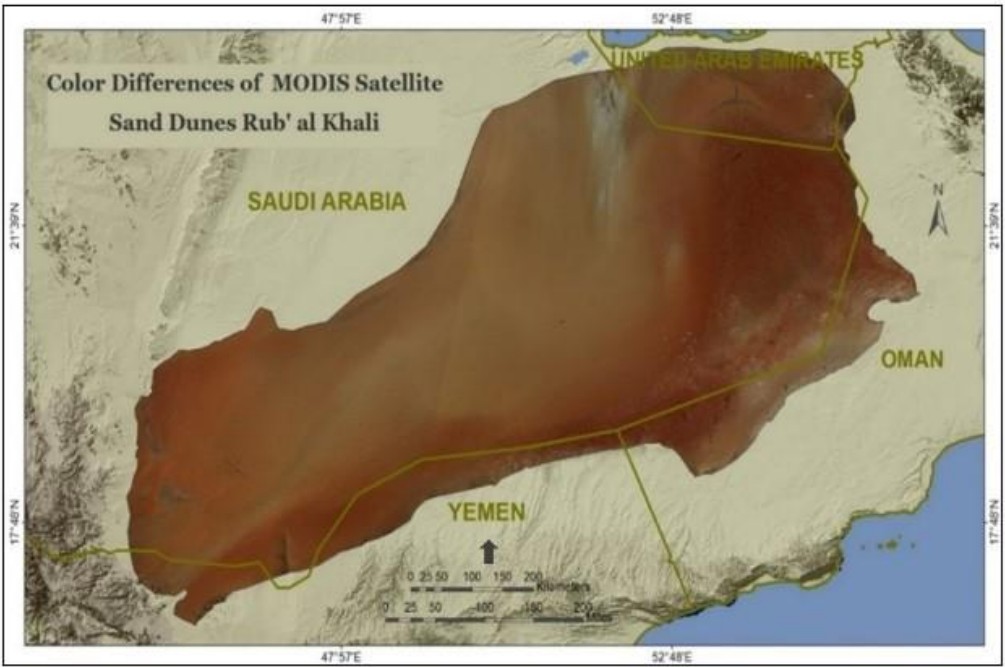

**Figure 6.** MODIS imagery showing variations in the color of the sand surface.

In the fourth step, the derivation of a drainage network Once the file was prepared, the DEM was analyzed using the geoprocessing tools contained in the Hydrology toolset (Fill Sinks, Flow Direction, Flow Accumulation) of the Spatial Analyst extension in ArcGIS. This procedure can explain the relationship between the white sand in the center of the Rub' al Khali and the drainage network in the Arabian Peninsula.

## 3. Results

The results of this work are very encouraging to estimate the sand volume for each sand dunes color of determent potential sources within the Rub' al Khali. The volume of the dune fields is an important variable in the numerical simulations used to study dune field dynamics [25]. To map the distribution of sand volume, the elevation of the desert floor beneath the dunes can be subtracted from the elevation of the dune surface. Conceptually, this process is rather straightforward, but creating a map of the desert floor beneath the dunes represents a challenging problem. In this analysis, several different methods are developed to map the desert floor beneath the dunes.

Figure 7 shows the surface of the desert floor underneath the sand dunes for four different analysis methods. Comparing the four maps, the overall pattern of elevation contours is similar, although there are noticeable differences in detail. While all four maps capture the eastward slope of the Rub' al Khali basin, the elevation surfaces created

from three interpolation methods are much smoother, and the surface created using the minimum focal statistics is far more detailed.

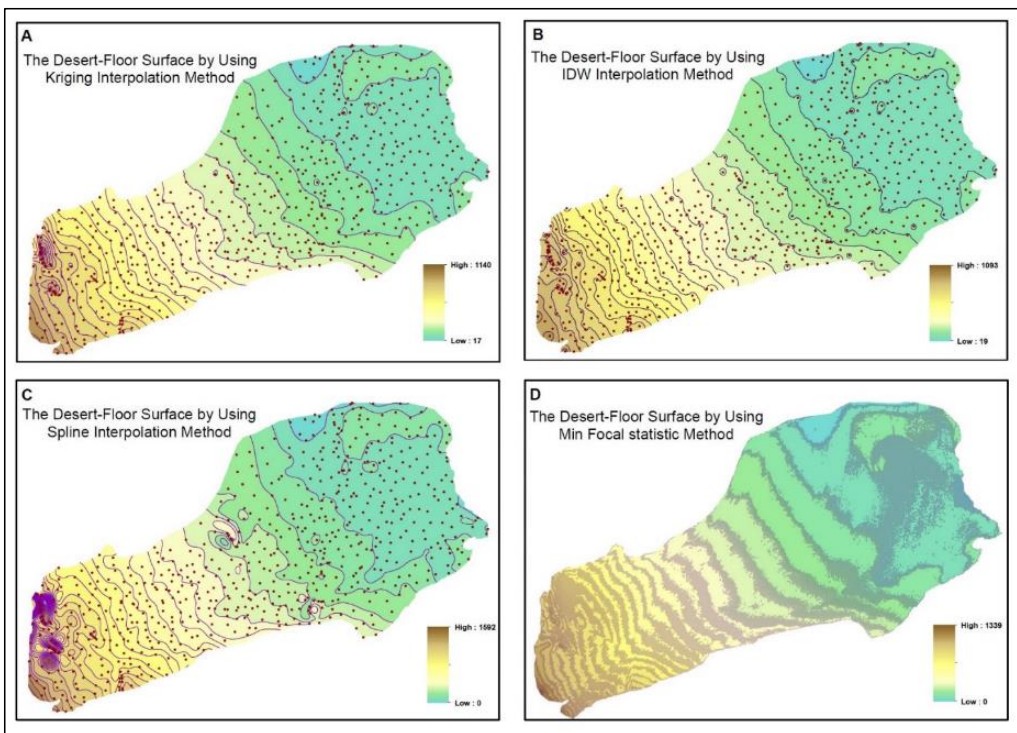

**Figure 7.** Desert floor elevation surfaces derived by (**A**) kriging, (**B**) IDW, (**C**) spline, and (**D**) minimum focal statistic.

The next step is to estimate the volume of the sand dune in the Rub' al Khali. While the desert is recognized as the largest sand sea in the world, there does not appear to be any published estimate of total sand volume, and there is certainly no map showing the spatial distribution of sand volume throughout the dune field. Some parts of the dune field contain massive dunes representing a huge volume of sand. In other parts of the dune field, the sand dunes are much smaller.

Table 2 shows the results of the volume analysis, expressed in trillions of cubic meters (thousands of billions). Comparing the volume calculations in this table, it becomes obvious that the three volumes derived by interpolating the desert floor from point elevations are quite similar, ranging in value from 7718 billion $m^3$ for IDW to 8261 billion $m^3$ for the spline. The difference between these two values is only 6.6%. Moreover, the volume derived from the kriging interpolation, 7903 billion $m^3$, is midway between the outside estimates. The focal statistic method shows a volume of 11,010 billion $m^3$ which is higher than the other discussed methods that will be discussed in the following pages.

**Table 2.** Comparison of the total sand volume in the Rub' al Khali calculated using four different methods to map the elevation surface of the desert floor.

| Volume Units | Kriging Volume | IDW Volume | Spline Volume | Focal Statistic Volume |
|---|---|---|---|---|
| Billion $m^3$ | 7903 | 7718 | 8261 | 11,010 |

Furthermore, the image mosaic of the Rub' al Khali was configured in a true color composite. In this mosaic, there are obvious differences in the color of the sand surface. The eastern part of the dune field appears as a noticeably dark brown when compared to the rest of the dune field. In a similar manner, the southern margin and western edge also

appear as a dark brown, although this brown color is slightly lighter. Equally important are the two major light-colored intrusions, one emerging from the northern margin of the dune field and one emerging from the southwest corner of the dune field. In particular, this light-colored intrusion from the southwestern edge of the dune field seems to extend all of the way into the interior of the dune field for almost 800 km. Analyzing differences in the sand surface of the dune field using satellite imagery is difficult owing to the similarity in the texture of the surface and the nearly homogeneous spectral response [7,26,27]. Figure 8 shows the results of the supervised classification in which the red sand is found mostly in the eastern part of the dune field and along the southern margin, with some areas of red sand found on the western edge of the dune field. In the eastern part of the dune field, the red sand class is interspersed with the sabkha class. Moving inward toward the center of the dune field, the red sand transitions to yellow sand, which then transitions to white sand. The white sand class covers a large area in the center of the dune field that extends to the northern margin. There is also a noticeable tail of white sand that extends to the far southwestern corner of the dune field. In a similar manner, yellow sand covers a large area in the western part of the dune field. The largest area classified as gypsum sand appears as an intrusion extending southward from the northern edge of the dune field. The gypsum class is also found along the southeastern margin of the dune field. In addition, Table 3 illustrates the areas of the supervised classification. The areas of dunes in the Rub' al Khali are mostly of yellow sand (~38%), white sand (~29%), and red sand (~15%), respectively. In contrast, the lowest areas of dune categories are rock sand and gypsum, estimated at 2.3% and 2.7%, respectively. The Sabkha area is also found in the southeastern of the dune field with 11%.

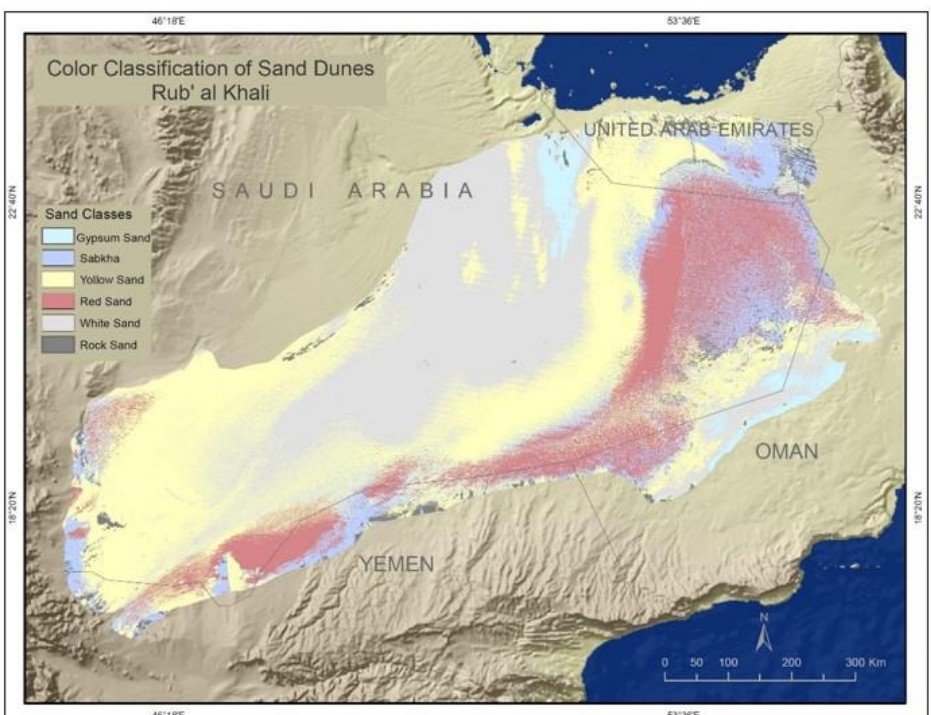

**Figure 8.** Supervised classification of MODIS imagery based on the analysis of true color bands.

To further investigate potential sources of sand, it is important to understand the drainage system in the region and how this drainage system appeared during the last glacial maximum. As noted earlier, during the last glacial maximum, sea level was much lower than it is today. During this time (~17,000–25,000 BP), the Arabian Gulf was dry and presumably exposed to northwest winds [28]. If this hypothesis is true, the dry surface of the gulf represents a very significant source of sand that was exposed over thousands of years. To better understand the hydrology of the region, in this analysis, a DEM is used

to reconstruct the paleo drainage system of the Arabian Peninsula when sea level was much lower.

**Table 3.** Areas of sand dune categories km$^2$.

| Categories | Area km$^2$ | % |
| --- | --- | --- |
| Sabkha | 59,532.34 | 11.62 |
| Yellow Sand | 195,694.31 | 38.20 |
| Gypsum Sand | 14,301.10 | 2.79 |
| Red Sand | 81,369.21 | 15.88 |
| Rock | 11,980.24 | 2.34 |
| White Sand | 149,404.02 | 29.16 |
| Total | 512,281.22 | 100.00 |

Figure 9 show the results of the drainage-network analysis. The map was created to show, (1) the very large size of the watershed covering 3,375,832 km$^2$ on the Arabian Peninsula, and (2) the shoreline of the Indian Ocean during the last glacial maximum when sea level was 120 m lower than today. During that time, the Arabian Gulf would have been dry, and the Tigris and Euphrates river system would transport sediment across the gulf basin to the modern Strait of Hormuz. Moreover, drainage from the Zagros Mountains in Iran and drainage from the Arabian Shield (Wadi Al-Rummah—Al-Batin, Wadi as Sabha, Wadi ad-Dawasir) would also contribute sediment to the gulf basin [29–31]. Clearly, with this large source of sediment exposed to Shamal (northwest) winds, the dry Arabian Gulf represents a very significant source area for the dune sand in the Rub' al Khali.

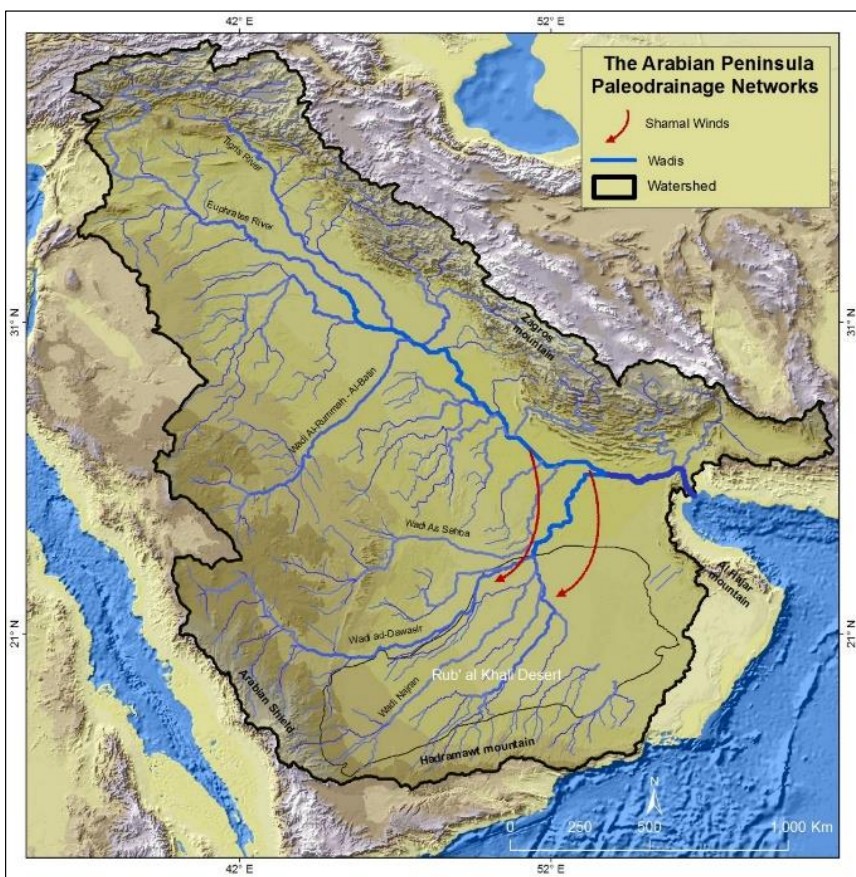

**Figure 9.** The paleo drainage network of the Arabian Peninsula derived from ETOPO1.

## 4. Discussion

The results from this study provide: (1) the first detailed map of the desert floor beneath the dunes, (2) the first reasonable estimate of the sand volume in the Rub' al Khali, (3) a detailed map of differences in sand color, (4) a map showing the paleo drainage of the region. Taken together, the results from the analyses in this study build upon previous research to enhance our understanding of the sand within the dune field and potential sand sources.

To calculate the volume of sand in each color class, the Zonal Statistics tool in ArcGIS was used to sum all of the cells in each class [20]. In addition, using a supervised classification, the analysis identified five unique signatures in the true color bands. These unique signatures are referred to as red sand, yellow sand, white sand, gypsum sand, and sand rock outcrops. Presumably, these differences in color represent differences in the mineral composition of the sand or the degree of oxidation. In either case, the color differences suggest differences in provenance or age. Table 4 and Figure 10 show the results of this volume analysis and each of the four different methods used to derive the desert floor surface. By volume, the dunes in the Rub' al Khali are mostly composed of yellow sand (~36%), red sand (~26%), and white sand (~20%). The analysis also showed a significant volume of sand associated with the sabkha color class (~15%), while the rock sand and gypsum were estimated at 1% and 2%, respectively. In this case, the kriging volume is 2.4% greater than IDW volume and 4.3% less than the spline volume. These results indicate that method used to interpolate the desert floor from point elevations does not greatly affect the final estimate of total sand volume.

**Table 4.** Sand volume estimate for each of the six color classes.

| Classes | Kriging Volume Billion m$^3$ | % | IDW Volume Billion m$^3$ | % | Spline Volume Billion m$^3$ | % | Focal Statistic Volume Billion m$^3$ | % |
|---|---|---|---|---|---|---|---|---|
| Sabkha | 1116 | 14 | 1163 | 15 | 1286 | 16 | 1558 | 14 |
| Yellow Sand | 2854 | 36 | 2596 | 34 | 2936 | 36 | 4120 | 37 |
| Gypsum Sand | 84 | 1 | 82 | 1 | 113 | 1 | 214 | 2 |
| Red Sand | 2154 | 27 | 2195 | 28 | 2130 | 26 | 2677 | 24 |
| Rock Sand | 146 | 2 | 161 | 2 | 143 | 2 | 262 | 2 |
| White Sand | 1549 | 20 | 1521 | 20 | 1652 | 20 | 2177 | 20 |
| Total | 7903 | 100 | 7718 | 100 | 8261 | 100 | 11,010 | 100 |

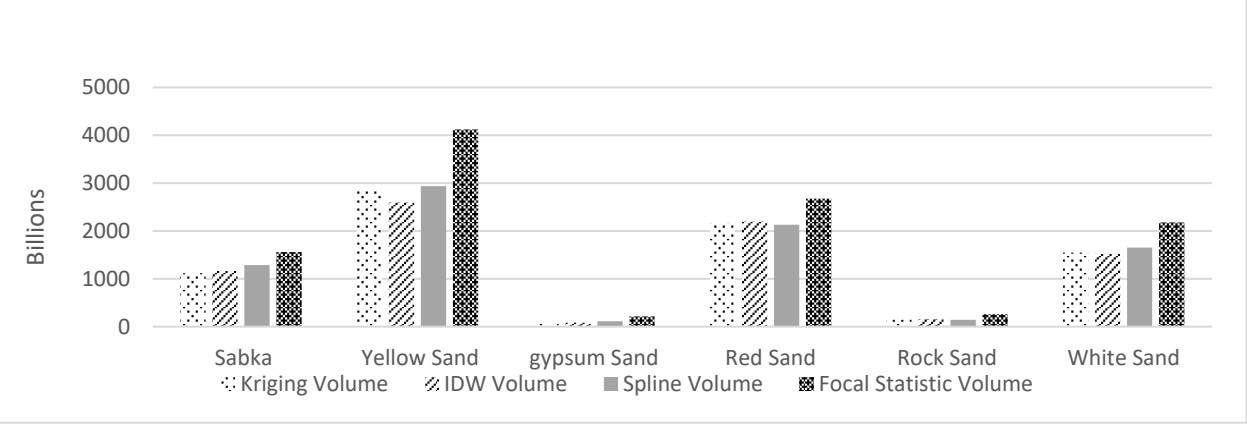

**Figure 10.** Comparison of sand volume calculations for each color class based on the four different methods used to map the desert floor surface (kriging, IDW, spline, and the minimum focal statistic).

In contrast to these three volume estimates, the volume of sand calculated using the minimum focal statistic to map the desert floor is significantly greater, 1101 billion m$^3$. This volume is 39.3% greater than the volume derived using the kriged desert surface (or the kriging volume is 28.2% less than the focal statistic volume).

Based on the foregoing analysis, it seems that the actual volume of sand in the Rub' al Khali falls within the range of about 8 to 11 trillion m$^3$. This is a tremendous volume of sand. Given that there are 1 billion m$^3$ in a km$^3$, this range in volume can also be expressed as 8000 to 11,000 km$^3$. It is also important to recall that these sand volume estimates were derived using a boundary that closely follows the boundary defined by the Saudi Geological Survey (2011) [10]. If the spatial extent of the Rub' al Khali is defined differently, presumably, the sand volume estimates would be slightly different.

While the results of this analysis are very promising, it remains unclear which of the four volume estimates is best or most accurate. Given that the complicated volume calculation procedure is very straightforward, the difference in volume estimates must be related to the derivation of the desert floor. In this regard, the analysis suggests that the desert floor elevation surface derived from focal statistic the minimum is, in general, lower than the desert floor surfaces derived from interpolation. With a lower desert floor elevation surface, the calculated volume is greater.

Knowing the volume or percentage of the volume associated with each color class, it now becomes meaningful to speculate on potential sources. Presumably, the relatively small volume of gypsum sand (~2%) found in the north and southeastern parts of the dune field is derived from local evaporative surfaces. The source or provenance of the other three major classes of sand is more difficult to define. Red sand (~26%) is associated with compound crescentic dunes (Megabarchans) in the eastern part of the dune field, the large linear dunes found along the southern margin of the dune field, and the large linear dunes found in the northwest corner of the dune field. While the analysis of the paleo drainage system suggests that the exposed Arabian Gulf is a major potential source, it seems likely that some of the red sand on the southern and western margins of the dune field is derived from local sources in the Hadramawt Arch to the south and the Sarawat Mountains in the west.

While the signature of the red sand is quite pronounced, the largest volume of sand (~36%) is associated with the yellow color class. Yellow sand covers most of the western part of the dunes field and seems to be a transitional color between red and white sand in the eastern part of the dune field. This suggests that the yellow sand might be derived from both local and regional sources, or it might be less oxidized, or it might be reworked, or it is composition might represent a combination of red and white sand.

While the provenance of yellow sand is difficult to evaluate, the spatial pattern of the white sand color class is very intriguing. White sand represents about 20% of the volume in the sand sea, and the sand is most closely associated with the smaller simple linear dunes in the center of the dune field. In addition, there is a large intrusion of white sand, which seems to emanate from a large wadi in the far southwestern corner of the dune field. This intrusion strongly suggests that white sand is derived from the Sarawat Mountains in the west. This observation supports the idea first proposed by [32], who suggested that a large percentage of the sand was derived from the western mountains when a wadi breached the escarpment of the western edge of the dune field. If this hypothesis is correct, it supports the idea that a large volume of white sand has been transported eastward through the wadi system into the central part of the dune field, where it is then subject to the northeast Shamal winds that blow the sand to the west.

The marked contrast in the spectral reflectance of the sand surface (excluding rock outcrops, sabkhas, and other interdune flats) strongly suggests that the dune field can be subdivided into four main regions that correspond to red sand, yellow sand, white sand, and gypsum sand. While it is not possible from this analysis of satellite data alone to determine whether these color differences are related to the composition of the sand or oxidation, the spatial pattern does spark interesting speculation. For example, the red

sand along the southern margin of the dune field might suggest a local source with wadis flowing northward from the Hadramawt-Dhofar Arch. At the same time, the red sand in the eastern part of the dune field might be related to oxidation, given the presence of water in the inter-dune sabkhas. Similarly, the long tail of white sand that extends to the southwestern corner of the dune field suggests that wadi sand might be carried hundreds of kilometers into the center of the dune field.

Additionally, within the Rub' al Khali, the dunes are obviously composed of sand from a number of different sources, but it is unclear how much of the sand is derived from the local mountains surrounding the basin and how much of the sand was transported into the basin from sources farther away. In the previous analysis of satellite imagery, there appear to be several distinct regions of sand based on differences in color. Presumably, these differences in color represent differences in the sand composition or degree of oxidation—both of which are relevant to understanding the history of the sand sea.

In the first case, rock outcrops, gravel sheets, sabkhas, and other non-sand inter-dune flats were identified and mapped as points using the zoom capabilities and very-high-resolution imagery available in Google Earth Pro shows several examples of these inter-dune areas that represent the desert floor beneath the dunes. Over a large area in the eastern part of the dune field, the sabkha surface is obvious between many of the compound crescentic dunes (megabarchans) and stars dunes. Similarly, over a large area covering the western part of the dune field, gravel sheets and other non-sand surfaces are evidently apparent between many of the large linear dunes. In the central part of the dune field, however, exposures of the desert floor are much more difficult to find. In this region of the dune field, the exposures often consist of small rock outcrops and small evaporative surfaces.

## 5. Conclusions

Understanding the provenance of the dune sand leads to many interesting questions. The results from this study seem to confirm previous ideas regarding the source of the sand, from both the local mountains and the exposed Arabian Gulf, but it is difficult to assess how this sand is distributed throughout the dune field. The analysis of dune color strongly suggests that the sand is not completely reworked and intermixed. If this is true, then a map of the spatial variability in the mineral composition of the sand could be very revealing. In this regard, the long tail of white sand emanating from the western mountains to the center of the dune field strongly suggests that local sources might be very important. While the analysis of the paleo drainage system suggests that the exposed Arabian Gulf is a major potential source, it seems likely that some of the red sand on the southern and western margins of the dune field is derived from local sources in the Hadramawt Arch to the south and the Sarawat Mountains in the west. In this regard, the analysis of the desert floor elevation surface derived from focal statistic the minimum is, in general, lower than the desert floor surfaces derived from interpolation. With a lower desert floor elevation surface, the calculated volume is greater. Clearly, with this large source of sediment exposed to Shamal (northwest) winds, the dry Arabian Gulf represents a very significant source area for the dune sand in the Rub' al Khali.

**Author Contributions:** Formal analysis, F.N.; Methodology, F.A.; Supervision, K.M.; Writing—original draft, F.N. All authors have read and agreed to the published version of the manuscript.

**Funding:** The authors extend their appreciation to the Deanship of Scientific Research at King Saud University for funding this work through Thesis Publication Fund No (TPF-010).

**Acknowledgments:** This author expresses his appreciation to the Deanship of Scientific Research at King Saud University and the Research Center at the Faculty of Arts for funding the current article.

**Conflicts of Interest:** The authors declare no conflict of interest.

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
