# Peer review of "Calculation of the Rub’ al Khali Sand Dune Volume for Estimating Potential Sand Sources"

_remotesensing, doi:10.3390/rs14051216_

Round 1

Reviewer 1 Report

The research issue is interesting, however, this reviewer wonders how to convince the audience about the reliability of the results and here are some issues are raised

  1. What is difference between local and regional sources of sand?
  2. How the authors deal with the big spatial resolution of different dataset (500, 90, and 1.8km)
  3. The four steps describe in the methodology section can be summarized into a infographic flowchart
  4. The ASTER dataset with 30 x 30 m resolution was not given in the Table 1.
  5. I don’t see the validation of the results and for the sand classification by using MODIS image there is no accuracy assessment.
  6. There is comparison of the estimated results by using four interpolation method, I suggest need to find further ways or census data to compare and validate the outcomes.

Author Response

Greetings,

responses to the first reviewer is attached.

thank you

Reviewer 2 Report

I have carefully read the paper and I believe that it is very interesting study worth publishing. This study concerns the evaluation and calculation of different colored sand of Rub’ al Khali desert on the southern Arabian Peninsula. In this interesting study, the authors used various geospatial technics, methods, and satellite image analysis in order to identify the different types of sand. This study brings interesting results to the field of aeolian geomorphology and other sciences which are interested in dunes.

For specific comments, please refer in the manuscript.

Overall, my suggestion is that the paper should be accepted after major revisions.

Author Response

Greetings

please find the comments responses attached

thank you

Reviewer 3 Report

The manuscript describes the different dune shapes and colors of sand, sabkha, and substratum outcrops of the Empty Quarter desert, Arabian Peninsula, through a geospatial technology approach, in particular analyzing satellite imagery dated from 6 and 12 years ago. Comparing DEMs, elaborated with different methods, and calculating the sand volume, the local and regional sand sources are identified.

The manuscript is almost well structured. The Introduction section needs some integrations, Materials and Methods are quite well explained. Results, Discussion, and Conclusions sections are congruent, sometimes with a few lacks. Figures and Tables are clear, but some integration in their caption and frame is necessary. References seem poor (31).

Comments and suggestions are listed hereinafter.

TEXT

1. Introduction, L26-56: Any information about the climate classification of the studied area is shown. This is useful to readers to better frame the geomorphic system. Consider citing the following articles:

Köppen W (1936) Das geographische System der Klimate. In: Köppen W and Geiger R (eds) Handbuch der Klimatologie. Berlin: Gerbru¨der Borntraeger, Vol. I, Part C, 44.
Kottek M, Grieser J, Beck C, et al. (2006) World map of the Köppen-Geiger climate classification updated. Meteorologische Zeitschrift 15: 259-263. Trewartha GT and Horn LH (1980) An Introduction to Climate. 5th ed. New York: McGraw Hill, 416.

A short geological-geomorphological outline should be added in the Introduction section for two reasons: 1, this could help not Arabian readers to understand the main outcrops and active/inactive processes of the studied area; 2, in the following sections there are indications about lithology and shapes, i.e., "Some of these interdune flats contain gravel sheets, sabkhas, and other apparently evaporative surfaces [14]." of L108-109. Finally, more worldwide case studies should be cited and the related references listed.

L35-36: "last glacial maximum (17,000 – 25,000 BP)" do you mean the Würm one? If yes, add it here, e.g., in the brackets before the interval time.

L36: "was 120 m" change to "was about 120 m", as the depth varies from 120 to 125 m lower than present-day sea level considering different authors. Any citation? Add at least one.

L73-75, L95-97: "For the true color band combination..." and "satellite imagery is analyzed to map the differences in the color of the sand surface, which is likely related to the composition and/or age of the sand." The description of colors is subjective and general, while it could be objective and more specific. About this issue, there is a lack in the manuscript. Soil colors may change due to mineralogic composition and grain size, but also for climate, the roughness of the stone rock or incoherent deposits, humidity content, sunlight orientation during the day, and so on. Consider associate to the true color map of Figure 7 or indicate in the main text the Munsell Table codes for the different categories of sand (yellow, gypsum, red, white, environment (sabkha), and substratum (rock)  in Table 3. Consider reading and citing the following publication:

Munsell AH (1975) Munsell Soil Color Charts. Munsell Color Macbeth, Division of Kollmorgen Corporation, Baltimore.

L137-145: "The three...smooth surface [15]." What of these interpolation methods apply the Simpson-Cavalieri method for 2D (and 3D) smooth surfaces? Consider that the S-C method is used also for monitoring and calculating loss and increase of sediment volume along incoherent morphologies profile in rapid change or flash remodeling due to an extreme event, e.g., sandy bars or coastal dunes, similarly to the scheme of Fig.4. Generally, kriging algorithms are reliable.

L161, Eq.1: "∑((dune surface elevation − desert floor elevation) ∗ grid cell area)" I suggest changing the first and last bracket to ∑ [ ( .... ) ... ] as usually in Math.

L189: "encouraging; To" is "encouraging to".

L197-198: "Mapping the desert floor beneath the dunes, the figure 6 shows the surface of the desert floor beneath the dunes ..." The sentence is redundant and confusing: rephrase it, and check the language.

L233: "The difference between these two values is only 6.6%" Usually, a good value resulting from such a comparison should be <5%: here, this is higher, and other values (unspecified in Table 2, but only in the Discussion, Table 4 and Figure 9) seem higher too and not negligible. Maybe, more information here could help readers to better understand differences.

L269: "km2" is "km2".

L275 and L287: " exposed to northwest winds [27]. ...  exposed to Shamal (northwest) winds, ..." In Figure 8 the direction of Shamal Winds (red arrows) is from NE to SW: these are not northwest winds but with a 45° rotation. Is this a typo in the text or a drawing error in the figure? Check and clarify this point. 

L303-305: "Presumably, these differences in color represent differences in the mineral composition of the sand or the degree of oxidation. In either case, the color differences suggest differences in provenance or age."

FIGURES

Figure 1, 5, 7, and 8 captions: specify the geographic coordinate system, e.g., WGS84 or other.

Figures 2 and 3: where are located these areas? Add geographic coordinates alongside the frame or - at least - at the two opposite vertexes, and indicate the coordinate system in the caption (see above the comment to Fig. 1 caption).

TABLES

Table 1 caption, L87: "Digital elevation models" change to "Digital Elevation Models" or simply DEMs, as introduced in L15.

Table 3,first row-first column: "categories" is "Categories"; first row-second column: "km2" is "km2".

Author Response

we have attached a document addressing your comments.

thank you

Round 2

Reviewer 1 Report

This paper is very intriguing about providing the spatial distribution, sand volume, and sand color of dune sand in the Rub’ al Khali desert based on DEM, and satellite imagery.

-        Furthermore, the author also explains clearly the source of different types of sands regarding local mountains, the exposed Arabian Gulf, the transportation from remote regions.

-        The results from the comparison among interpolation methods are useful for other researches in the same field. 

-        However, there are some rooms that authors could consider improving their papers:

(1)     The language is unclear, making it difficult to follow. The author should revise the language to improve readability.

(2)     The description of methodologies is clear; however, it is better to make a flowchart for summarizing the main concept.

(3)     The quality of figures should improve (e.g., size of the text in Figure 8) for followable.

Author Response

Dear reviewer, 

Your comments have been considered and changes have been applied.

Thank you

Reviewer 2 Report

The manuscript has been revised according to the suggestions. I believe that this paper should be published as it is.

Author Response

Thank you